# Effect of HTLV-1 Infection on the Clinical Course of Patients with Rheumatoid Arthritis

**DOI:** 10.3390/v14071460

**Published:** 2022-07-01

**Authors:** Kunihiko Umekita

**Affiliations:** Division of Respirology, Rheumatology, Infectious Diseases and Neurology, Internal Medicine, Faculty of Medicine, University of Miyazaki, 5200 Kihara, Kiyotake, Miyazaki 889-1692, Japan; kunihiko_umekita@med.miyazaki-u.ac.jp; Tel.: +81-985-85-7284

**Keywords:** human T-cell leukemia virus type 1, rheumatic diseases, rheumatoid arthritis, adult T-cell leukemia/lymphoma, HTLV-1-associated myelopathy/tropical spastic paraparesis, disease-modifying antirheumatic drugs

## Abstract

Human T-cell leukemia virus type 1 (HTLV-1) is the causative agent of adult T-cell leukemia/lymphoma (ATL) and HTLV-1-associated myelopathy/tropical spastic paraparesis (HAM/TSP). The effects of HTLV-1 on health are not fully elucidated. Epidemiological studies have shown that the prevalence of HTLV-1 infection is high in patients with rheumatic diseases. The prevalence of comorbidities, such as Sjögren’s syndrome and rheumatoid arthritis (RA), is higher in patients with HAM/TSP than the in general population. Studies have shown the effects of HTLV-1-infection on the clinical course of RA. Major questions on the association between HTLV-1 infection and RA: (1) Is it possible that HTLV-1 infection causes RA? (2) Do patients with RA who are infected with HTLV-1 have different clinical features? (3) Are immunosuppressants associated with an increased prevalence of HAM/TSP or ATL in RA patients with HTLV-1 infection? Is ATL an immunosuppressive therapy-associated lymphoproliferative disorder? No large-scale studies have investigated the incidence of ATL in patients with RA. However, several studies have reported the development of ATL in patients with RA who have HTLV-1 infection. This review aimed to shed light on the association between HTLV-1 infection and RA and summarize the unmet medical needs of RA patients with HTLV-1 infection.

## 1. Introduction

Human T-cell leukemia virus type 1 (HTLV-1) is a human retrovirus that is a causative agent of adult T-cell leukemia/lymphoma (ATL) and HTLV-1-associated myelopathy/tropical spastic paraparesis (HAM/TSP) [1,2,3,4]. HTLV-1 causes HTLV-1-associated uveitis and infective dermatitis, in addition to HAM/TSP and ATL [5,6,7]. The virus infects approximately 10 million people worldwide, with endemic foci in Japan, the Caribbean, South America, and Central Africa [8]. Numerous people are infected with HTLV-1 in developing countries. In Japan, there are about 1 million HTLV-1 carriers [8,9], which is the highest worldwide [8]. HTLV-1 carriers have a lifetime risk of 2–7% [9,10] and 0.25–3.8% for developing ATL and HAM/TSP, respectively [11]. A recent study showed novel evidence that at least 4000 adolescents and adults (of whom 77% are women) acquire new HTLV-1 infection via horizontal transmission annually in Japan. Several studies have reported that the new HTLV-1 infection rates are highest in women aged 50–59 years and in men aged 60–69 years in both endemic and non-endemic regions. Meanwhile, the HTLV-1 infection rate in young men aged 20–29 years is higher in non-endemic regions (including metropolitan areas) than that in endemic regions [12]. The main target of HTLV-1 infection is a cluster of differentiation (CD) 4-positive T-lymphocytes. Infection of these cells with HTLV-1 reportedly alters their function. The alteration of T-lymphocyte functions may be involved in the pathogenesis of not only HTLV-1-associated disease but also some inflammatory disorders, such as bronchiectasis and polymyositis [13,14]. Individuals with HTLV-1 infection are at a higher risk of infectious diseases (such as strongyloidiasis and tuberculosis) [15,16]. There have been several reports about the relationship between HTLV-1 infection and rheumatic diseases, but the results remain inconclusive [14,15]. Rheumatoid arthritis (RA), defined as chronic inflammation of the joints and bone destruction, is the most common rheumatic disease. Smoking, gingivitis, and Epstein–Barr virus (EBV) infection have been considered environmental risk factors, particularly in relation to the production of autoantibodies [17]. Genetic factors, such as certain types of *Human Leukocyte Antigen–**DR* isotype genes and polymorphisms of multiple genes, also have been considered important for its etiology [18,19]. Anti-citrullinated protein antibodies (ACPAs), which are one of the antibodies in RA, may be involved in the pathogenesis of RA. ACPA-positive RA patients have shown aggressive bone destruction. The response to antirheumatic therapies tends to be inadequate in ACPA-positive RA compared with that in ACPA-negative RA. To maintain the chronic inflammation of RA, cell–cell interactions and interactions with cytokines among lymphocytes, macrophages, and synovial fibroblasts (SFs) in the joints have crucial roles. Recently, disease-modifying antirheumatic drugs (DMARDs), including the targets of these cell–cell interactions, and cytokines have been developed and have dramatically improved the clinical course of RA [20]. Biologics and molecular-targeted synthetic DMARDs have revolutionized the clinical course and outcome of patients with RA [20]. However, there is still a risk posed by biologics and synthetic DMARDs for developing serious infections and malignancies in patients with RA [21,22]. For example, EBV reactivation has been considered to be one of the causes of immunosuppressive and immunomodulatory therapy-associated lymphoproliferative disorders (LPDs) in patients with RA [23,24].

The major research and clinical questions about HTLV-1 infection and rheumatic diseases reviewed in this article are as follows: (1) Is it possible that HTLV-1 infection is one of the causes of rheumatic diseases? (2) Do patients with rheumatic diseases display any different clinical features when they are HTLV-1 carriers? (3) Do immunosuppressants, including biologics and molecular-targeted DMARDs, increase the prevalence of HAM/TSP or ATL in RA patients with HTLV-1 infection? Should development of ATL be considered to be an immunosuppressive therapy-associated LPD? A search strategy was used to select articles on the association between HTLV-1 infection and RA, including case series, clinical research, cohort study, and meta-analysis, in PubMed. All studies were published from 1980 to 2022. The key words used to search the articles were as follows: HTLV-1, ATL, rheumatic manifestations, arthropathy, arthritis, RA, antirheumatic therapies, lymphoproliferative disorders, and HAM/TSP. This review aimed to provide novel information on the association between HTLV infection and RA and to summarize the unmet medical needs of RA patients with HTLV-1 infection in routine clinical practice.

## 2. HTLV-1 and Rheumatic Diseases

It remains unclear whether HTLV-1 infection etiologically contributes to the development of RA. Several epidemiological studies have shown that the prevalence of HTLV-1 infection is higher in patients with rheumatic diseases, such as RA, polymyositis, and Sjögren’s syndrome, than in healthy controls, such as blood donors [25,26,27]. A recent single-cohort study indicated that the HTLV-1 prevalence was significantly higher in RA participants than in non-RA participants [28]. After adjustment for age, sex, and hepatitis C virus infection, HTLV-1 was significantly associated with prevalent RA [28]. A systematic review of epidemiological studies showed that HTLV-1 infection was associated with an elevated risk of RA and Sjögren’s syndrome among rheumatic diseases [16]. These results may suggest the effect of HTLV-1 infection in the etiology of rheumatic diseases; however, HTLV-1-positive patients comprise only a minor proportion of patients with rheumatic diseases, even in the most prevalent areas of HTLV-1. HTLV-1-positive patients comprised only 6% of patients with RA in our cohort in Miyazaki, Japan, which is one of the most endemic areas for HTLV-1 [29]. A certain proportion of HTLV-1-positive patients with arthropathy were reported to exhibit mono- or oligo-arthritis of the large joints. [30]. Biopsy samples from their synovial tissues tested positive for HTLV-1. In the 1990s, the concept of HTLV-1-associated arthropathy (HAAP) was proposed [31,32], although it remains unclear whether HAAP differs from HTLV-1-positive RA. Table 1 shows the differences in clinical manifestations between HAAP and RA. Onset seems to occur in more elderly patients in HAAP than in RA. The clinical features of affected joints differ between HAAP and RA. Mono- or oligo-arthropathy was present in patients with HAAP. In addition, radiologic studies have revealed that the destruction of bone seemed to be relatively milder in HAAP than in RA [32]. In the histopathological findings of synovial tissues, the infiltration of HTLV-1-infected cells seem to be observed in not only HAAP, but also RA if HTLV-1-infected. ACPA is an autoantibody that defines the clinical features of RA. Contrasting genetic backgrounds in ACPA-positive and ACPA-negative RA support the notion that these are in fact two distinct disease subsets, with different underlying pathogeneses [33]. Several studies have indicated that the clinical features and laboratory data, including the prevalence of rheumatoid factor and ACPA, are similar between HTLV-1-positive and HTLV-1-negative RA patients [29,34,35]. No studies have investigated the prevalence of ACPA in not only HAAP, but also in HTLV-1 carriers without rheumatic manifestations. Therefore, it is difficult to conclude that HTLV-1 infection alone causes RA, although it has still not been determined if HTLV-1 infection is a causative agent for arthropathy or poly-arthritis, especially in patients who are seronegative for these autoantibodies.

## 3. HTLV-1 Causes Inflammation In Vitro

HTLV-1 primarily infects CD4+T-lymphocytes and is thought to alter their functions and lineages. It is possible that certain clones of HTLV-1-infected cells proliferate and alter both innate and acquired immunity of HTLV-1 carriers. Previous studies have suggested that the proinflammatory status of HTLV-1-infected cells is characterized by the presence of cytokines, such as TNF, IFN-γ, IL-6, IP-10, TGF, and GM-CSF [36,37]. Both the proliferation and cytokines production of RASFs was increased by coculture with these HTLV-1-infected cells [38]. Chronic inflammatory diseases, including arthritis, have been shown to develop in transgenic mice with the HTLV-1 *Tax* and *HBZ* genes [39,40]. Tax, an HTLV-1-related protein, is an activator of NF-kB [41,42,43]. Expressions of mRNAs for the proinflammatory cytokines IL-1 and IL-6 have been observed in synovial tissues of arthropathic *Tax*-transgenic (Tg) mice but not in those of healthy Tg mice [44]. In the *Tax*-Tg mice, TNF-alpha expression had not been observed in either arthropathic or healthy Tg mouse synovial tissues mRNA [44]. Exocrinopathy resembling Sjögren’s syndrome also has been reported in HTLV-1 *Tax-Tg* mice [45]. Compared with HTLV-1-negative patients, HTLV-1-positive patients with Sjögren’s syndrome have been reported to have uveitis and lung diseases more frequently but to have fewer anti-nuclear antibodies [46,47]. These characteristics are more evident in HTLV-1-positive patients with Sjögren’s syndrome, which is associated with HAM/TSP, suggesting a relationship among these diseases [46,47]. According to these results, HTLV-1 Tax may have roles in the inflammation of HTLV-1 carriers. On the other hand, regarding the role of HTLV-1 HBZ, IFN-γ promotes inflammation and development of T-cell lymphoma in *HBZ*-Tg mice [48]. IL-6 is well known as a proinflammatory cytokine and has important roles in the pathogenesis of rheumatic diseases. Interestingly, in mice crossbred from IL-6 knockout mice and HBZ-Tg mice, loss of IL-6 accelerates inflammation and lymphomagenesis in HBZ-Tg mice [49]. IL-6 innately inhibits regulatory T-cell differentiation, suggesting that IL-6 functions as a suppressor against HBZ-associated complications. HBZ upregulates expression of the immunosuppressive cytokine IL-10. IL-10 promotes T-cell proliferation only in the presence of HBZ. As a mechanism of growth promotion by IL-10, HBZ interacts with STAT1 and STAT3 and modulates the IL-10/JAK/STAT signaling pathway [49]. HBZ activates the STAT-responsive elements, such as IFN-stimulated response element and IFN-γ activation site, which are suppressed by IL-10. These findings suggest that HTLV-1 promotes the proliferation of infected T cells by hijacking the machinery of regulatory T-cell differentiation. IL-10 induced by HBZ probably suppresses the host immune response and concurrently promotes the proliferation of HTLV-1-infected T cells. Both HTLV-1 Tax and HBZ may be involved in the inflammatory responses and ATL progression [50].

HAM/TSP is a chronic inflammatory disease of the central nervous system that has high levels of HTLV-1 proviral load (PVL) and polyclonal expansion of HTLV-1-infected cells. In one study, peripheral blood mononuclear cells (PBMCs) isolated from patients with HAM/TSP showed autonomously produced inflammatory cytokines, such as IFN-γ, IL-6, and TNF-alpha [51]. In further studies, HTLV-1 Tax was shown to activate the *t-bet* gene with reduced expression of FoxP3 in the infected cells, resulting in their differentiation toward Th1 balance in HAM/TSP [52,53,54]. The production of chemokines in cultured peripheral blood mononuclear cells obtained from patients with HAM/TSP was shown to be increased in another study [55]. CD4+CD25+CCR4+T-lymphocytes in HAM/TSP produce IFN-γ, activate astrocytes in the central nervous system with CXCL10 expression, and induce migration of Th1-like T-lymphocytes into the central nervous system [56]. Both HTLV-1 carriers and patients with HAM/TSP have been reported to be associated with various chronic inflammatory diseases, including rheumatic diseases [16,57,58,59,60].

In the inflammation sites of RA, HTLV-1-infected cells may have important roles in the exacerbation of inflammation. Figure 1 shows a hypothetical schema of worsening inflammation in HTLV-1-positive RA. The number of HTLV-1-infected cells has been reported to increase not only in the peripheral blood but also in the synovial fluid of patients with RA [60], although the roles of these HTLV-1-infected cells in the pathogenesis of RA have not yet been revealed. Ex vivo cultures of lymphocytes from HTLV-1 carriers show spontaneous proliferation [61], which has also been observed more clearly in patients with HAM/TSP [62,63]. Production of TNF and IFN-γ was observed in cultured PBMCs obtained from patients with HAM/TSP [54,64]. If a process similar to that of HAM/TSP occurs in HTLV-1-positive RA patients, HTLV-1 infection can be an environmental factor responsible for initiation and/or maintenance of chronic inflammation in rheumatic diseases. Actually, PBMCs obtained from HTLV-1-positive RA patients have been shown to be autonomously involved in IFN-γ production ex vivo [65]. In one study, the SFs obtained from RA patients co-cultured with an HTLV-1-infected cell line were found to secrete inflammatory cytokines [27]. However, further clarification is necessary to determine if HTLV-1-infected T-lymphocytes in patients with RA show characteristics that resemble HTLV-1-infected cells in ATL or HAM/TSP. Dysregulations of the balance between functionally opposite cytokines, such as IFN-γ/IL-10 balance, are thought to contribute to the pathogenesis of HTLV-1 infection [66]. Thus, an analysis of the function of HTLV-1-infected T-lymphocytes and cytokines in rheumatic diseases is necessary.

## 4. Are There Any Differences in Clinical Features between HTLV-1-Positive and Negative RA Patients

### 4.1. Disease Activity and Clinical Response in HTLV-1-Positive RA

The levels of C-reactive protein (CRP), which is a marker of RA activity, have been shown to be higher in HTLV-1-positive patients with RA before receiving treatment with biologic DMARDs than the levels of CRP HTLV-1-negative patients [67]. In addition, a recent cohort study suggested that the patient pain visual analog scale, global assessment, and health assessment questionnaire scores were higher in HTLV-1-positive RA patients than in HTLV-1-negative RA patients, even though there were no differences in inflammatory markers between them [68]. These results suggest that worse pain and physical disability were commonly observed in HTLV-1-positive RA patients. Several reports have suggested that rheumatologic manifestations, such as arthralgia and lower back pain, have been observed in HTLV1 carriers [69,70]. Complaints of pain seem to be common among HTLV-1-infected individuals, regardless of neurological manifestations associated with HAM/TSP [71]. An epidemiologic systematic review and meta-analysis indicated an association between HTLV-1 infection and rheumatologic disorders, such as fibromyalgia and arthropathy, in HTLV-1 carriers [16]. Although the pathogenesis of pain associated with HTLV-1 infection has not been elucidated, HTLV-1 infection may contribute to the poorer health assessment of HTLV-1 carriers. Two retrospective observational studies have shown the attenuated effectiveness of TNF inhibitors in HTLV-1-positive patients with RA [34,67]. Several studies have reported that the response to TNF inhibitors was better in ACPA-negative RA patients than ACPA-positive RA patients [72]. In HTLV-1-positive RA patients, the efficacy of TNF inhibitors was reported to be inadequate in not only ACPA-positive but also ACPA-negative RA patients [34]. These data suggested that HTLV-1 infection induces more RA-related inflammation and contributes to the attenuated effectiveness of TNF inhibitors, although its mechanism is not clear. Another important question is whether or not HTLV-1-positive patients with RA have resistance not only to TNF inhibitors but also to other DMARDs. Initiation of tocilizumab or abatacept as first biologics may be effective in both HTLV-1-positive and -negative RA patients [35]. It is possible that IL-6 and activated T cells may be more predominant in the inflammatory responses of HTLV-1-positive RA patients than in HTLV-1-negative RA patients. However, it remains unclear whether non-TNF biologics are safe and effective for HTLV-1-positive RA patients.

### 4.2. High Incidence of Serious Infections in HTLV-1-Positive RA Patients

HTLV-1 infection may alter the host immune responses against *Mycobacterium tuberculosis* (TB), *Strongyloides stercoralis* (*S. stercoralis*), EBV, cytomegalovirus, and *Pneumocystis jirovejii* [73,74]. A higher prevalence of TB and *S. stercoralis* has been observed in HTLV-1 carriers than in healthy subjects without HTLV-1 infection [16]. During antirheumatic therapies, the prevalence risk of these opportunistic infections has also increased in patients with RA. Some reports have found that the tuberculin skin test reaction in HTLV-1-positive individuals was attenuated relative to that in HTLV-1-negative individuals [75,76]. However, it remains unclear whether HTLV-1-infection increases the incidence of these infections in HTLV-1-positive RA patients.

Hashiba et al. reported a case of HTLV-1-positive RA patients who developed *S. stercoralis* colitis after administration of a TNF inhibitor [77]. Recently, a HTLV-1 RA cohort study found a higher incidence of serious infections requiring hospitalization in HTLV-1-positive RA patients than in HTLV-1-negative RA patients [68]. In this study, the most frequently observed complications were serious respiratory infections in HTLV-1-positive RA patients. Respiratory comorbidities are well-known risk factors for developing respiratory infections in RA patients. Several reports have demonstrated that the incidence of respiratory disorders, such as bronchitis and bronchiectasis, is higher in HTLV-1-infected individuals than in healthy individuals [13,78]. A meta-analysis has also indicated an association between HTLV-1 infection and respiratory complications [13,16]. Atsumi et al. [79] suggested that HTLV-1 infection is an independent risk factor for community-acquired pneumonia (CAP) and that HTLV-1-infected patients tended to develop a relatively severe form of pneumonia. One reason why CAP develops in HTLV1-infected patients could be the structural changes in the lungs. A previous report found structural changes in the lungs on computed tomography images in 30.1% of HTLV-1 carriers [80]. Therefore, HTLV-1-positive RA patients may be at increased risk for developing serious respiratory infections.

### 4.3. Invalid T-SPOT.TB, a Screening Test for Latent TB Infection

The incidence of TB is reported to be higher in RA patients receiving anti-TNF therapy compared with the general population [81,82]. National recommendations for latent TB infection (LTBI) screening based on patient medical history, clinical examination, tuberculin skin testing (TST), and chest radiographs have been effective in reducing TB incidence [83]. IFN-γ release assays (IGRAs) have been established as a useful screening test for LTBI. In the general population, IGRAs are more effective than TST for diagnosing active TB infection or LTBI [84]. The Centers for Disease Control and Prevention updated the guidelines for recommending IGRAs to detect TB infection [85]. In daily practice of rheumatic diseases, IGRAs are reported to be useful for diagnosing LTBI before the initiation of biologic therapy, such as anti-TNF agents [86]. However, it is possible that HTLV-1-infection affects the T-SPOT.TB result, which is an IGRA. A recent study demonstrated that approximately 55% of the HTLV-1-positive RA patients showed invalid T-SPOT.TB (Oxford Immunotec Ltd., Abingdon, UK) assay results owing to a spot count of >10 in the negative controls. HTLV-1 PVL values seemed to be significantly higher in patients with invalid results than in those without invalid results [65]. HTLV-1-infected T cells often behave in a manner similar to T helper (Th) 1-like cells, which autonomously produce IFN-γ [52,66]. HTLV-1-infected T cells reportedly show autonomous proliferation and produce inflammatory cytokines, such as IL-6, TNF-alpha, and IFN-γ [66]. HTLV-1 infection alters the original function of T cells via HTLV-1-associated proteins, such as Tax and HBZ [66]. In HTLV-1-positive RA patients, these IFN-γ producing cells, which are suspected to be HTLV-1-infected T cells, may be responsible for the invalid T-SPOT.TB assay results. Therefore, T-SPOT.TB assay results should be interpreted with caution when screening for LTBI in HTLV-1-positive RA patients. In the future, it is necessary to research and develop an LTBI screening test in HTLV-1-positive RA patients.

## 5. Is the Incidence of HAM/TSP and ATL High in HTLV-1-Positive RA Patients

It is unknown if the association of rheumatic diseases and treatment against such diseases affects the natural clinical course of HTLV-1 infection, particularly in view of the development of HTLV-1-associated diseases, such as HAM/TSP and ATL.

### 5.1. High Incidence of RA in HAM/TSP Patients

Patients with RA are considered to have a Th1- and Th17-dominant immune status. The immune status of HAM/TSP is also considered Th1 dominant. Therefore, whether or not RA affects the etiology and clinical condition of HAM/TSP is an important question. It is also important to know if the incidence of HAM/TSP in patients with RA who have been an HTLV-1 carrier or who acquired HTLV-1 infection in adulthood is higher than that in asymptomatic HTLV-1 carriers. Indeed, the prevalence of comorbidities of Sjögren’s syndrome and RA in patients with HAM/TSP was recently reported to be as high as 3.7% and 2.7%, respectively [87]. This prevalence of RA appears to be much higher in patients with HAM/TSP than in the general population. However, there is no evidence of the prevalence of HAM/TSP in HTLV-1-positive RA patients. Given the rarity of HAM/TSP, conducting large-scale prospective observational studies to clarify this issue has been challenging. Several cohort studies of HTLV-1-positive RA Japanese patients with short observational periods suggest that any new onset of HAM/TSP was not observed during antirheumatic therapies, including biologics [29,68,88].

We must also consider the effect of the DMARDs that patients with RA are receiving. Indeed, worsening symptoms of both HAM/TSP and HTLV-1-associated uveitis were reported in an HTLV-1-positive patient with RA who received treatment with a soluble IL-6 receptor inhibitor (sIL-6Ri) [89]. Conversely, a case report indicated that treatment with sIL-6Ri improved the disease activity of RA patients with HAM/TSP without a flare of neurologic manifestations [90]. Given the difference in clinical outcomes between these cases, it remains unclear whether sIL-6Ri is safe for RA patients with HAM/TSP. From a rheumatology point of view, it is necessary to determine if antirheumatic therapies, including biologics and targeted synthetic DMARDs, are safe and effective for HAM/TSP patients with RA. We need more data on HTLV-1-positive patients with rheumatic diseases and their treatment.

### 5.2. Do Immunosuppressive Therapies Increase the Incidence of ATL in HTLV-1-Positive RA Patients

Patients with RA reportedly have higher incidence rates of malignant lymphoma, although the surface phenotype of lymphoma found in RA is B-cell predominant [91]. In addition, the incidence of LPDs, including lymphoma, has been reported to increase in RA patients who were treated with MTX [92]. LPDs that are related to treatment with immunosuppressive drugs are categorized as “other iatrogenic immunodeficiency-associated LPDs” [93]. Most of the LPD cases reported are B-cell type or Hodgkin’s disease, and the involvement of EBV infection in these LPD cases is suspected. For ATL, a higher PVL, advanced age, family history of ATL, and first opportunity for HTLV-1 testing during treatment for other diseases were reported as independent risk factors for the progression of ATL in HTLV-1 carriers [94]. Further studies revealed that accumulated somatic mutations were involved in the ATL progression [95]. Since these somatic mutations cause attenuated Tax expression and increase PD-L1 expression, ATL cells, including its precursor cells, could escape anti-viral and the host’s anti-tumor immunities. Moreover, some somatic mutations may also drive the tumorigenesis of HTLV-1-infected cells to ATL progression. Clonally expanded HTLV-1-infected cells have these high-risk somatic mutations. Therefore, HTLV-1-positive RA patients who have these risk factors may be categorized as high-risk RA patients for developing ATL.

It is unclear whether DMARD treatment on RA must be recognized as an important risk factor for the development of ATL in HTLV-1-positive RA patients. MTX is well known as the anchor drug for RA and has immunosuppressive effects caused by inhibition of nucleic acid synthesis. Another DMARD, tacrolimus, specifically suppresses T-cell function by inhibiting calcineurin. Biologics and targeted synthetic DMARDs mainly inhibit the function of proinflammatory cytokines and immune response cells. It is unknown if there are any direct effects of these medicines on HTLV-1-infected cells in vivo; however, it is clear that these medicines cause various levels of immune suppression against anti-viral immunity. Several reports have suggested that antirheumatic therapies increased the incidence of viral reactivation, such as EBV, hepatitis B virus, and varicella-zoster virus. Since ATL could be considered to be one of the virus-associated LPDs, suppression of anti-HTLV-1 immunity may be involved in the progression of ATL (Figure 2). Indeed, clonal expansion of HTLV-1-infected T-lymphocytes in HTLV-1 carriers was considered to be controlled by the pressure of an immune surveillance system for a long time [96]. A clone that progressed to ATL was reportedly found in the peripheral blood even 8 years before the onset of ATL [97]. An HTLV-1 Tax-specific cytotoxic T-lymphocyte response has been reported in asymptomatic HTLV-1 carriers [98]. For example, a patient has been found to develop ATL after treatment with R-CHOP chemotherapy (rituximab, cyclophosphamide, doxorubicin, vincristine, and prednisone) against B-cell lymphoma-associated hemophagocytic syndrome [99]. Another study reported a high incidence of ATL in liver transplantation recipients who received tacrolimus as an immunosuppressant [100]. In addition, a recent study found a high risk of HTLV-1 transmission and HAM/TSP development after a short incubation period in HTLV-1–negative recipients of kidney transplants from HTLV-1–positive donors [101]. Kidney transplantation from a positive donor to a negative recipient carries a high risk of infection. Because the HTLV-1-negative recipients were immunosuppressed after receiving an immunosuppressant, the high risk of HAM/TSP development only in HTLV-1–negative recipients receiving a transplant from a positive donor may indicate that a lack of anti-HTLV-1 immunity is an important risk factor for HAM/TSP after receipt of a kidney transplant from a positive donor. These reports suggested that lack of anti-HTLV-1 immunity may be involved in the progression of ATL.

In daily clinical practice of RA, several cases of ATL have been reported in patients with rheumatic diseases during treatment with DMARDs, including MTX [102,103,104,105,106,107]. MTX-EVB-associated LPDs often show spontaneous regression after cessation of MTX [92]. This phenomenon has also been reported in cases of ATL in RA patients treated with MTX [105,106]. Conversely, reports observing HTLV-1-positive RA patients for several years showed that ATL cases are rarely identified [29,88,108]. Treatment with DMARDs, including both MTX and biologics, did not significantly increase HTLV-1 PVLs in HTLV-1-positive patients with RA [29]. However, the cessation of MTX treatment decreased HTLV-1 PVLs in HTLV-1-positive RA patients with a high PVL in the same report [29]. It remains uncertain whether treatment with DMARDs contributes to the development of ATL in HTLV-1-positive RA patients. There is a possibility that immunosuppression caused by DMARDs promotes the development of ATL, particularly if a patient with RA already has clones of HTLV-1-infected cells with a nature similar to that of ATL cells. Because no data have been available on the analysis of clonal expansion of HTLV-1 infected cells in patients with RA, further studies are necessary on this topic.

## 6. Conclusions

Considering the rarity of HTLV-1 positivity in RA, it may be challenging to answer the following clinical questions: Is HTLV-1 an etiologic factor of RA? Do the clinical characteristics of RA patients with HTLV-1 infection differ from those who did not? Is there evidence showing that immunosuppressants are associated with an increased incidence of ATL in RA patients with HTLV-1 infection (Figure 3)? The prevalence of HTLV-1 infection is two or three times higher in participants with RA than in those without RA [28]. However, the prevalence rate of HTLV-1-antibody positivity was found to only be 6% in patients with RA in an endemic area [29]. The population of RA patients with HTLV1 infection was small in this HTLV-1 endemic area. Therefore, whether HTLV-1 is an etiologic agent of RA remains unclear. This review showed several differences in terms of the clinical characteristics between RA patients with HTLV-1 infection and those who did not. The efficacy and safety of biologic DMARDs may differ between patients with RA who have HTLV-1 infection and those who do not. Regarding the safety of biologics, a case report indicated that IL-6 inhibitor exacerbated the neurologic characteristics in HAM/TSP patient with RA. In addition, patients with RA who have HTLV-1 infection may be at a high risk of infection requiring hospitalization during antirheumatic therapies. HTLV-1 infection could be an environmental factor affecting the clinical features of patients with RA. Finally, whether immunosuppressive agents, including biologics and targeted synthetic DMARDs, can affect ATL progression remains unclear. This review included several case reports. However, there was no direct-evidence indicating the association between the development of HTLV-1-related disorders and the use of immunosuppressants. Several factors affecting ATL progression, including host immune senescence, attenuation of anti-HTLV-1 immunity, and accumulation of somatic mutation in HTLV-1-infected cells, should be considered. In relation to anti-HTLV-1 immunity suppression, immunosuppressive agents may be involved in the pathogenesis of ATL. However, this clinical question is challenging to answer based on data from few cohort studies. The questions raised in this article are relevant to countries other than Japan, and it is necessary to collect global data on HTLV-1 infection in rheumatic diseases. Rheumatologists must determine if they should pay special attention to a patient with rheumatic disease who have HTLV-1 infection. Moreover, the importance of HTLV-1 testing before the initiation of treatment with DMARDs must be determined by rheumatologists. Patients with RA who have HTLV-1 infection may develop HTLV-1-associated disorders, such as ATL, HAM/TSP, and HU. Thus, rheumatologists should collaborate with hematological, neurological, and ophthalmological specialists. However, to date, there are no recommendations for the management of these unmet medical needs based on robust evidence. Hence, in the future, further long-term, large-scale, nationwide studies on RA patients with HTLV-1 infection must be conducted.

## Figures and Tables

**Figure 1 viruses-14-01460-f001:**
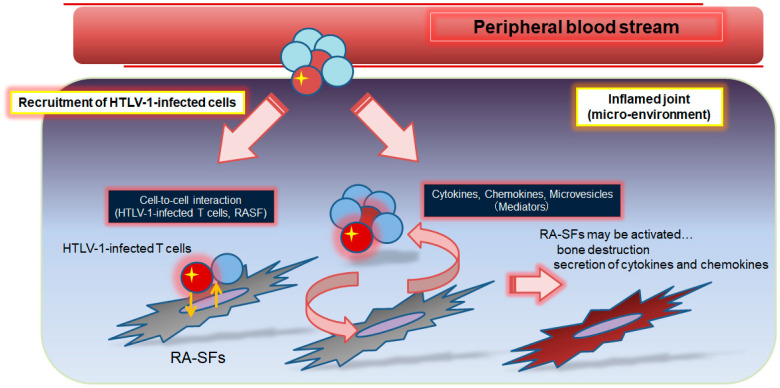
**Hypothesis of the pathogenesis in the worsening of inflammation in HTLV-1-positive patients with rheumatoid arthritis (RA).** Several reports have indicated that HTLV-1-infected T cells infiltrate the inflamed joints of patients with RA. These HTLV-1-infected T cells reportedly exhibit autonomous proliferation and release cytokines and chemokines via HTLV-1-associated proteins, such as Tax and HBZ. Therefore, it is possible that there are interactions between HTLV-1-infected T cells and RA synovial fibroblasts (RA-SFs).

**Figure 2 viruses-14-01460-f002:**
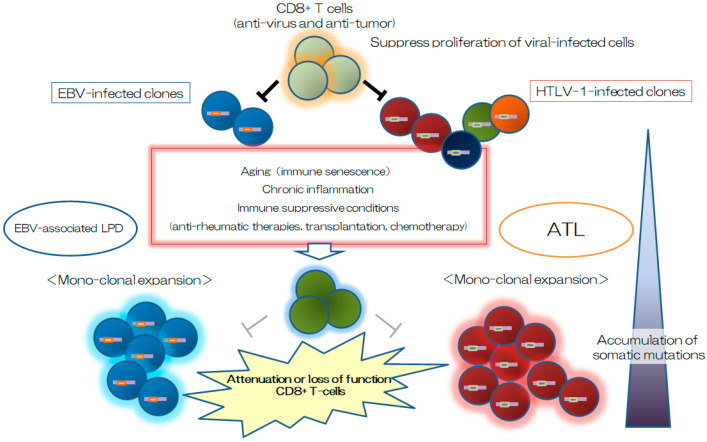
**Involvement of host immunity status in the pathogenesis of viral-associated lymphoproliferative disorder (LPD).** It is unclear whether there any underlying common causes between Epstein–Barr virus (EBV)-associated LPD and adult T-cell leukemia (ATL). Host immunity has an important role in suppressing expansion of viral-infected cells in a host. Immunosuppressive treatment, including not only antirheumatic therapies, but also organ transplantation, attenuates the anti-viral immunity of the host. Furthermore, aging, immune senescence, and inflammation may also cause remodeling of the immune system, such as innate immunity and acquired immunity. Indeed, it is well known that the incidence of EBV-associated LPD was increased by methotrexate during antirheumatic therapy. On the other hand, accumulation of somatic mutations is involved in the clonal expansion of ATL cells because ATL cells that accumulate somatic mutations can escape from the anti-viral and anti-tumor immune systems. Although HTLV-1-positive patients with rheumatic diseases may be categorized as having a high risk for developing ATL, it remains unclear whether antirheumatic therapies and chronic inflammation increase the incidence of ATL.

**Figure 3 viruses-14-01460-f003:**
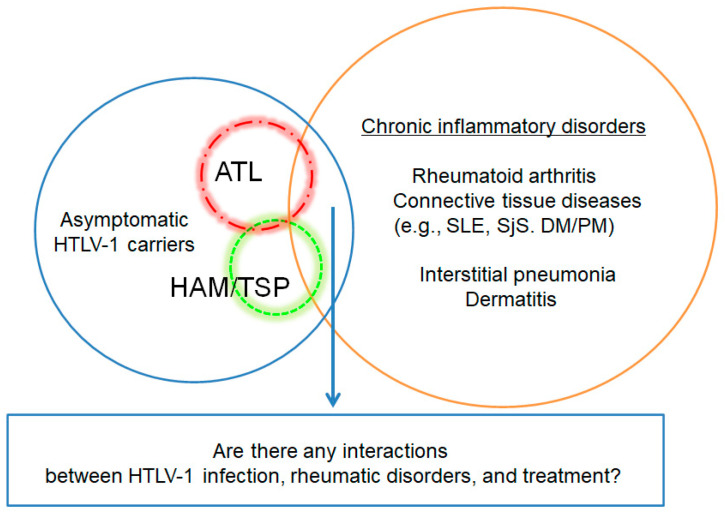
**Human T-cell leukemia virus type 1 (HTLV-1) infection in chronic inflammatory disorders.** Given the rarity of HTLV-1-positive patients with chronic inflammatory disorders, such as rheumatoid arthritis and connective tissue diseases, it may be difficult to determine if there are any interactions between HTLV-1 infection, rheumatic disorders, and treatment. However, several studies have suggested that the clinical manifestations of HTLV-1-positive patients with RA may differ from those of HTLV-1-negative patients with RA. In addition, some patients have been reported to develop ATL during immunosuppressive treatment. Therefore, many clinical and research questions about these issues remain.

**Table 1 viruses-14-01460-t001:** Comparison of clinical manifestations between patients with HTLV-1-associated arthropathy and those with rheumatoid arthritis.

	HTLV-1-Associated Arthropathy (HAAP)	Rheumatoid Arthritis
**Sex**	Unknown	Female > male
**Age at onset**	>60 years-old	40–50 years old
**HTLV-1 antibody**	Required	Not required
**Affected joints**	Mono-, oligo-arthropathy Large joints, predominant	Poly-arthritis
**Destruction of joints**	Rare or mild	Progressive
**Positivity of RF**	Sometimes	Positive
**Positivity of ACPAs**	No available data	Positive
**Increase in CRP**	None~mild	Increase
**Histopathological findings of synovial tissues**		
**Proliferation**	Mild	Severe
**Erosions**	Rare	Severe
**Infiltration of HTLV-1-infected T-cells**	Yes	Yes (If HTLV-1-infected)
**Treatment**	NSAIDs, corticosteroids, IFN-alpha	DMARDs

HTLV-1: human T-cell leukemia virus type 1, RF: rheumatoid factor, ACPAs: anti-citrullinated protein antibodies, CRP: C-reactive protein, NSAIDs: non-steroidal anti-inflammatory drugs, IFN: interferon, DMARDs: disease modifying antirheumatic drugs.

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
