# Peer review of "Effect of HTLV-1 Infection on the Clinical Course of Patients with Rheumatoid Arthritis"

_viruses, 2022, doi:10.3390/v14071460_

Round 1

Reviewer 1 Report

The manuscript submitted by Umekita R. is a comprehensive review of the effects of HTLV infection on the clinical course of rheumatoid arthritis. The author states that he addresses the questions of whether HTLV can be considered an etiologic agent of rheumatoid arthritis, whether infection affects the clinical presentation of rheumatoid arthritis, and finally, whether treatment with immunosuppressants, including biologics and molecular- targeted DMARDs, can increase the incidence of ATL.

Several considerations need to be made about the manuscript:

The objectives of the study and the methodology used are not clearly stated. The author intends to review three key questions: a possible causal relationship between HTLV and rheumatoid arthritis, a change in the clinical course of rheumatoid arthritis in infected individuals, and whether there is a higher incidence of ATL in HTLV-infected patients treated with immunosuppressants. I suggest that the author restate the goal for the association between HTLV and rheumatoid arthritis. The treatment issue can be addressed as a secondary issue in the article.

1) Clearly state the objectives of the manuscript in the abstract.

2) In the introduction of the article, the author should point out that HTLV-1 is known to cause HTLV-associated uveitis and infective dermatitis in children, in addition to HAM / TSP and ATL. The virus is also associated with the development of various inflammatory changes such as bronchiectasis and poliomyositis. People infected with HTLV are at higher risk for infectious diseases (strongyloidiasis, tuberculosis, and others). It is important to emphasize that HTLV is associated with the occurrence of various alterations and other diseases, unlike previously thought.

3) Provide epidemiologic data on HTLV infection, estimated number of infected persons, geographic distribution, and demographic data (higher prevalence in women, many infected persons in developing countries).

 4) The author must indicate the search strategy used to select the articles, e.g., keywords, search bases, period evaluated, inclusion and exclusion criteria of the articles.

5) The conclusion of the article must be restated as it is confusing. The author needs to clearly summarize the results found in accordance with the questions proposed in the review. Can HTLV be considered an etiologic agent of rheumatoid arthritis? Do the clinical manifestations in HTLV-infected patients differ from patients without HTLV? Is there evidence that immunosuppressants increase the incidence of ATL in patients with HTLV?

6) What are the author's recommendations for professionals caring for patients with HTLV and suspected rheumatoid arthritis?

Reviewer 2 Report

Suggestion to the author:
To present the study with a detailed description of the methodology used to review the literature. Otherwise it is not possible to evaluate this study

Reviewer 3 Report

The review “Effect of HTLV-1 infection on the clinical course of patients with rheumatoid arthritis” by Umeika summarizes the current knowledge of HTLV-1 and rheumatic disease correlations. The review discusses three significant issues: HTLV-infection correlation to rheumatic diseases; the clinical feature of HTLV-1 positive subjects with rheumatic disease; the possible effect the immunosuppressive therapies in the development of HTLV-1 associate diseases, mainly HAM/TSP and ATL: The contribution of the literature has been sufficiently summarized and discussed with proper explanation.

Therefore, I think that the paper is suitable for publication

A few comments are listed below:

-       The abstract describes the topic, but it will also be helpful if it describes the organization/structure of the review.

-       In section 3, Line  7: “Tax, which is one of the HTLV-1 related proteins, was reported” “ requires the insertion of some references such as doi: 10.1042/BSR20211921.,doi:10.3389/fmicb.2018.00285,doi:10.1111/febs.14492.

-       Is it more common to name “biological and synthetic DMARD” instead of “biologics and synthetic DMARD?.

-       - Figures 1, 2, and 3: It might be clearer to put all captions at the bottom instead of the title at the top and commentary below the figures.

Round 2

Reviewer 1 Report

The author needs to clearly indicate the keywords used (the way the keywords were presented in the text resulted in zero articles when entered into PubMed). I suggest that the author includes a methodology section in the manuscript describing the step-by-step process of bibliographic review, article selection, included and excluded articles, etc. This can be recorded in a flow chart.

Reviewer 2 Report

This is the author-reviewed version of the article. The author included  methodology the study and made a considerable revision of the abstract and the text, making the manuscript clearer and more adequate for publication.